# Unsupervised Control through Non-Parametric Discriminative Rewards

## Abstract

Learning to control an environment without hand-crafted rewards or expert data remains challenging and is at the frontier of reinforcement learning research. We present an unsupervised learning algorithm to train agents to achieve perceptually-specified goals using only a stream of observations and actions. Our agent simultaneously learns a goal-conditioned policy and a goal achievement reward function that measures how similar a state is to the goal state. This dual optimization leads to a co-operative game, giving rise to a learned reward function that reflects similarity in controllable aspects of the environment instead of distance in the space of observations. We demonstrate the efficacy of our agent to learn, in an unsupervised manner, to reach a diverse set of goals on three domains – Atari, the DeepMind Control Suite and DeepMind Lab.

## 1 Introduction

Currently, the best performing methods on many reinforcement learning benchmark problems combine model-free reinforcement learning methods with policies represented using deep neural networks (Horgan et al., 2018; Espeholt et al., 2018). Despite reaching or surpassing human-level performance on many challenging tasks, deep model-free reinforcement learning methods that learn purely from the reward signal learn in a way that differs greatly from the manner in which humans learn. In the case of learning to play a video game, a human player not only acquires a strategy for achieving a high score, but also gains a degree of mastery of the environment in the process. Notably, a human player quickly learns which aspects of the environment are under their control as well as how to control them, as evidenced by their ability to rapidly adapt to novel reward functions (Lake et al., 2017).

Focusing learning on mastery of the environment instead of optimizing a single scalar reward function has many potential benefits. One benefit is that learning is possible even in the absence of an extrinsic reward signal or with an extrinsic reward signal that is very sparse. Another benefit is that an agent that has fully mastered its environment should be able to reach arbitrary achievable goals, which would allow it to generalize to tasks on which it wasn't explicitly trained. Building reinforcement learning agents that aim for environment mastery instead of or in addition to learning about a scalar reward signal is currently an open challenge.

One way to represent such knowledge about an environment is using an environment model. Model-based reinforcement learning methods aim to learn accurate environment models and use them either for planning or for training a policy. While learning accurate environment models of some visually rich environments is now possible (Oh et al., 2015; Chiappa et al., 2018; Ha & Schmidhuber, 2018) using learned models in model-based reinforcement learning has proved to be challenging and model-free approaches still dominate common benchmarks.

We present a new model-free agent architecture of Discriminative Embedding Reward Networks, or DISCERN for short. DISCERN learns to control an environment in an unsupervised way by learning purely from the stream of observations and actions. The aim of our agent is to learn a goal-conditioned policy $\pi_\theta(a|s; s_g)$ (Kaelbling, 1993; Schaul et al., 2015) which can reach any goal state $s_g$ that is reachable from the current state $s$. We show how to learn a goal achievement reward function $r(s; s_g)$ that measures how similar state $s$ is to state $s_g$ using a mutual information objective at the same time as learning $\pi_\theta(a|s; s_g)$. The resulting learned reward function $r(s; s_g)$ measures similarity in the space of controllable aspects of the environment instead of in the space of raw observations.

Crucially, the DISCERN architecture is able to deal with goal states that are not perfectly reachable, for example, due to the presence of distractor objects that are not under the agent's control. In such cases the goal-conditioned policy learned by DISCERN tends to seek states where the controllable elements match those in the goal state as closely as possible.

We demonstrate the effectiveness of our approach on three domains – Atari games, continuous control tasks from the DeepMind Control Suite, and DeepMind Lab. We show that our agent learns to successfully achieve a wide variety of visually-specified goals, discovering underlying degrees of controllability of an environment in a purely unsupervised manner and without access to an extrinsic reward signal.

## 2   PROBLEM FORMULATION

In the standard reinforcement learning setup an agent interacts with an environment over discrete time steps. At each time step $t$ the agent observes the current state $s_t$ and selects an action $a_t$ according to a policy $\pi(a_t|s_t)$. The agent then receives a reward $r_t = r(s_t, a_t)$ and transitions to the next state $s_{t+1}$. The aim of learning is to maximize the expected discounted return $R = \sum_{t=0}^{\infty} \gamma^t r_t$ of policy $\pi$ where $\gamma \in [0, 1)$ is a discount factor.

In this work we focus on learning only from the stream of actions and observations in order to forego the need for an extrinsic reward function. Motivated by the idea that an agent capable of reaching any reachable goal state $s_g$ from the current state $s$ has complete mastery of its environment, we pose the problem of learning in the absence of rewards as one of learning a goal-conditioned policy $\pi_\theta(a|s; s_g)$ with parameters $\theta$. More specifically, we assume that the agent interacts with an environment defined by a transition distribution $p(s_{t+1}|s_t, a_t)$. We define a goal-reaching problem as follows. At the beginning of each episode, the agent receives a goal $s_g$ sampled from a distribution over possible goals $p_{goal}$. For example, $p_{goal}$ could be the uniform distribution over all previously visited states. The agent then acts for $T$ steps according to the goal-conditioned policy $\pi_\theta(a|s; s_g)$ receiving a reward of 0 for each of the first $T - 1$ actions and a reward of $r(s_T; s_g)$ after the last action, where $r(s; s_g) \in [0, 1]$ for all $s$ and $s_g$ [1]. The goal achievement reward function $r(s; s_g)$ measures the degree to which being in state $s$ achieves goal $s_g$. The episode terminates upon the agent receiving the reward $r(s_T; s_g)$ and a new episode begins.

It is straightforward to train $\pi_\theta(a|s; s_g)$ in a tabular environment using the indicator reward $r(s; s_g) = \mathbb{1}\{s = s_g\}$. We are, however, interested in environments with continuous high-dimensional observation spaces. While there is extensive prior work on learning goal-conditioned policies (Kaelbling, 1993; Schaul et al., 2015; Andrychowicz et al., 2017; Held et al., 2017; Pathak et al., 2018), the reward function is often hand-crafted, limiting generality of the approaches. In the few cases where the reward is learned, the learning objective is typically tied to a pre-specified notion of visual similarity. Learning to achieve goals based purely on visual similarity is unlikely to work in complex, real world environments due to the possible variations in appearance of objects, or goal-irrelevant perceptual context. We now turn to the problem of learning a goal achievement reward function $r_\phi(s; s_g)$ with parameters $\phi$ for high-dimensional state spaces.

## 3   LEARNING A REWARD FUNCTION BY MAXIMIZING MUTUAL INFORMATION

We aim to simultaneously learn a goal-conditioned policy $\pi_\theta$ and a goal achievement reward function $r_\phi$ by maximizing the mutual information between the goal state $s_g$ and the achieved state $s_T$ as shown in (1).

$$I(s_g, s_T) = H(s_g) + \mathbb{E}_{s_g, s_T \sim p(s_g, s_T)} \log p(s_g|s_T) \tag{1}$$

Note that we are slightly overloading notation by treating $s_g$ as a random variable distributed according to $p_{goal}$. Similarly, $s_T$ is a random variable distributed according to the state distribution induced by running $\pi_\theta$ for $T$ steps for goal states sampled from $p_{goal}$.

The prior work of Gregor et al. (2016) showed how to learn a set of abstract options by optimizing a similar objective, namely the mutual information between an abstract option and the achieved

---

[1]More generally the time budget $T$ for achieving a goal need not be fixed and could either depend on the goal state and the initial environment state, or be determined by the agent itself.

state. Following their approach, we simplify (1) in two ways. First, we rewrite the expectation in terms of the goal distribution $p_{goal}$ and the goal conditioned policy $\pi_\theta$. Second, we lower bound the expectation term by replacing $p(s_g|s_T)$ with a variational distribution $q_\phi(s_g|s_T)$ with parameters $\phi$ following Barber & Agakov (2004), leading to

$$I(s_g, s_T) \geq H(s_g) + \mathbb{E}_{s_g \sim p_{goal}, s_1, \dots s_T \sim \pi_\theta(\cdots|s_g)} \log q_\phi(s_g|s_T). \tag{2}$$

Finally, we discard the entropy term $H(s_g)$ from (2) because it does not depend on either the policy parameters $\theta$ or the variational distribution parameters $\phi$, giving our overall objective

$$O_{\text{DISCERN}} = \mathbb{E}_{s_g \sim p_{goal}, s_1, \dots s_T \sim \pi_\theta(\cdots|s_g)} \log q_\phi(s_g|s_T). \tag{3}$$

This objective may seem difficult to work with because the variational distribution $q_\phi$ is a distribution over possible goals $s_g$, which in our case are high-dimensional observations, such as images. We sidestep the difficulty of directly modelling the density of high-dimensional observations by restricting the set of possible goals to be a finite subset of previously encountered states that evolves over time (Lin, 1993). Restricting the support of $q_\phi$ to a finite set of goals turns the problem of learning $q_\phi$ into a problem of modelling the conditional distribution of possible intended goals given an achieved state, which obviates the requirement of modelling arbitrary statistical dependencies in the observations.[2]

**Optimization:** The expectation in the DISCERN objective is with respect to the distribution of trajectories generated by the goal-conditioned policy $\pi_\theta$ acting in the environment against goals drawn from the goal distribution $p_{goal}$. We can therefore optimize this objective with respect to policy parameters $\theta$ by repeatedly generating trajectories and performing reinforcement learning updates on $\pi_\theta$ with a reward of $\log q_\phi(s_g|s_T)$ given at time $T$ and 0 for other time steps. Optimizing the objective with respect to the variational distribution parameters $\phi$ is also straightforward since it is equivalent to a maximum likelihood classification objective. As will be discussed in the next section, we found that using a reward that is a non-linear transformation mapping $\log q_\phi(s_g|s_T)$ to $[0, 1]$ worked better in practice. Nevertheless, since the reward for the goal conditioned-policy is a function of $\log q_\phi(s_g|s_T)$, training the variational distribution function $q_\phi$ amounts to learning a reward function.

**Communication Game Interpretation:** Dual optimization of the DISCERN objective has an appealing interpretation as a cooperative communication game between two players – an *imitator* that corresponds to the goal-conditioned policy and a *teacher* that corresponds to the variational distribution. At the beginning of each round or episode of the game the imitator is provided with a goal state. The aim of the imitator is to communicate the goal state to the teacher by taking $T$ actions in the environment. After the imitator takes $T$ actions, the teacher has to guess which state from a set of possible goals was given to the imitator purely from observing the final state $s_T$ reached by the imitator. The teacher does this by assigning a probability to each candidate goal state that it was the goal given to the imitator at the start of the episode, i.e. it produces a distribution $p(s_g|s_T)$. The objective of both players is for the teacher to guess the goal given to the imitator correctly as measured by the log probability assigned by the teacher to the correct goal.

## 4 DISCRIMINATIVE EMBEDDING REWARD NETWORKS

We now describe the DISCERN algorithm – a practical instantiation of the approach for jointly learning $\pi_\theta(a|s; s_g)$ and $r(s; s_g)$ outlined in the previous section.

**Goal distribution:** We adopt a non-parametric approach to the problem of proposing goals, whereby we maintain a fixed size buffer $\mathcal{G}$ of past observations from which we sample goals during training. We update $\mathcal{G}$ by replacing the contents of an existing buffer slot with an observation from the agent's recent experience according to some substitution strategy; in this work we considered two such strategies, detailed in Appendix A3. This means that the space of goals available for training drifts as a function of the agent's experience, and states which may not have been reachable under a poorly trained policy become reachable and available for substitution into the goal buffer, leading to a naturally induced curriculum. In this work, we sample training goals for our agent uniformly at

---

[2]See e.g. Lafferty et al. (2001) for a discussion of the merits of modelling a restricted conditional distribution rather than a joint distribution when given the choice.

random from the goal buffer, leaving the incorporation of more explicitly instantiated curricula to future work.

**Goal achievement reward:** We train a goal achievement reward function $r(s; s_g)$ used to compute rewards for the goal-conditioned policy based on a learned measure of state similarity. We parameterize $r(s; s_g)$ as the positive part of the cosine similarity between $s$ and $s_g$ in a learned embedding space, although shaping functions other than rectification could be explored. The state embedding in which we measure cosine similarity is the composition of a feature transformation $h(\cdot)$ and a learned $L^2$-normalized mapping $\xi_\phi(\cdot)$. In our implementation, where states and goals are represented as 2-D RGB images, we take $h(\cdot)$ to be the final layer features of the convolutional network learned by the policy in order to avoid learning a second convolutional network. We find this works well provided that while training $r$, we treat $h(\cdot)$ as fixed and do not adapt the convolutional network's parameters with respect to the reward learner's loss. This has the effect of regularizing the reward learner by limiting its adaptive capacity while avoiding the need to introduce a hyperparameter weighing the two losses against one another.

We train $\xi_\phi(\cdot)$ according to a *goal-discrimination* objective suggested by (3). However, rather than using the set of all goals in the buffer $\mathcal{G}$ as the set of possible classes in the goal discriminator, we sample a small subset for each trajectory. Specifically, the set of possible classes includes the goal $g$ for the trajectory and $K$ *decoy* observations $d_1, d_2, \ldots, d_K$ from the same distribution as $s_g$. Letting

$$\ell_g = \xi_\phi(h(s_T))^\mathsf{T} \xi_\phi(h(g)) \tag{4}$$

we maximize the log likelihood given by

$$\log \hat{q}(s_g = g | s_T; d_1, \ldots d_K, \pi_\theta) = \log \frac{\exp\left(\beta \ell_g\right)}{\exp\left(\beta \ell_g\right) + \sum_{k=1}^{K} \exp\left(\beta \xi_\phi(h(s_T))^\mathsf{T} \xi_\phi(h(d_k))\right)} \tag{5}$$

where $\beta$ is an inverse temperature hyperparameter which we fix to $K + 1$ in all experiments. Note that (5) is a maximum log likelihood training objective for a softmax nearest neighbour classifier in a learned embedding space, making it similar to a matching network (Vinyals et al., 2016). Intuitively, updating the embedding $\xi_\phi$ using the objective in (5) aims to increase the cosine similarity between $e(s_T)$ and $e(g)$ and to decrease the cosine similarity between $e(s_T)$ and the decoy embeddings $e(d), \ldots, e(d_K)$. Subsampling the set of possible classes as we do is a known method for approximate maximum likelihood training of a softmax classifier with many classes (Bengio & Sénécal, 2003).

We use $\max(0, \ell_g)$ as the reward for reaching state $s_T$ when given goal $g$. We found that this reward function is better behaved than the reward $\log \hat{q}(s_g = g | s_T; d_1, \ldots d_K, \pi_\theta)$ suggested by the DISCERN objective in Section 3 since it is scaled to lie in $[0, 1]$. The reward we use is also less noisy since, unlike $\log \hat{q}$, it does not depend on the decoy states.

**Goal-conditioned policy:** The goal-conditioned policy $\pi_\theta(a|s; s_g)$ is trained to optimize the goal achievement reward $r(s; s_g)$. In this paper, $\pi_\theta(a|s; s_g)$ is an $\epsilon$-greedy policy of a goal-conditioned action-value function Q with parameters $\theta$. Q is trained using Q-learning and minibatch experience replay; specifically, we use the variant of $Q(\lambda)$ due to Peng (see Chapter 7, Sutton & Barto (1998)).

**Goal relabelling:** We use a form of goal relabelling (Kaelbling, 1993) or hindsight experience replay (Andrychowicz et al., 2017; Nair & Hinton, 2006) as a source successfully achieved goals as well as to regularize the embedding $e(\cdot)$. Specifically, for the purposes of parameter updates (in both the policy *and* the reward learner) we substitute, with probability $p_{\text{HER}}$ the goal with an observation selected from the final $H$ steps of the trajectory, and consider the agent to have received a reward of 1. The motivation, in the case of the policy, is similar to that of previous work, i.e. that being in state $s_t$ should correspond to having achieved the goal of reaching $s_t$. When employed in the reward learner, it amounts to encouraging temporally consistent state embeddings (Mobahi et al., 2009; Sermanet et al., 2017), i.e. encouraging observations which are nearby in time to have similar embeddings.

Pseudocode for the DISCERN algorithm, decomposed into an experience-gathering (possibly distributed) actor process and a centralized learner process, is given in Algorithm 1.

## 5 RELATED WORK

The problem of reinforcement learning in the context of multiple goals dates at least to Kaelbling (1993), where the problem was examined in the context of grid worlds where the state space is

---

**Algorithm 1:** DISCERN

---

**procedure** ACTOR

  **Input** : Time budget $T$, policy parameters $\theta$, goal embedding parameters $\phi$, shared goal
           buffer $\mathcal{G}$, hindsight replay window $H$, hindsight replay rate $p_{\text{HER}}$

  **repeat**

    $\hat{\pi}_\theta \leftarrow$ BEHAVIOR-POLICY$(\theta)$                                         /* e.g. $\epsilon$-greedy */

    $g \sim \mathcal{G}$

    $r_{1:T-1} \leftarrow 0$

    **for** $t \leftarrow 1 \ldots T$ **do**

      Take action $a_t \sim \hat{\pi}_\theta(s_t; g)$ obtaining $s_{t+1}$ from $p(s_{t+1}|s_t, a_t)$

      $\mathcal{G} \leftarrow$ PROPOSE-GOAL-SUBSTITUTION$(\mathcal{G}, s_t)$            /* See Appendix A3 */

    **end**

    **with** probability $p_{\text{HER}}$,

      Sample $s_{\text{HER}}$ uniformly from $\{s_{T-H}, \ldots, s_T\}$ and set $g \leftarrow s_{\text{HER}}, r_T \leftarrow 1$

    **otherwise**

      Compute $\ell_g$ using (4)

      $r_T \leftarrow \max(0, \ell_g)$

    Send $(s_{1:T}, a_{1:T}, r_{1:T}, g)$ to the learner.

    Poll the learner periodically for updated values of $\theta, \phi$.

    Reset the environment if the episode has terminated.

  **until** *termination*

**procedure** LEARNER

  **Input** : Batch size $B$, number of decoys $K$, initial policy parameters $\theta$, initial goal
            embedding parameters $\phi$

  **repeat**

    Assemble batch of experience $\mathcal{B} = \{(s_{1:T}^b, a_{1:T}^b, r_{1:T}^b, g^b)\}_{b=1}^B$

    **for** $b \leftarrow 1 \ldots B$ **do**

      Sample $K$ decoy goals $d_1^b, d_2^b, \ldots, d_K^b \sim \mathcal{G}$

    **end**

    Use an off-policy reinforcement learning algorithm to update $\theta$ based on $\mathcal{B}$

    Update $\phi$ to maximize $\frac{1}{B}\sum_{b=1}^B \log \hat{q}(s_g = g^b | s_T; d_1, \ldots d_K, \pi_\theta)$ computed by (5)

  **until** *termination*

---

small and enumerable. Sutton et al. (2011) proposed generalized value functions (GVFs) as a way of representing knowledge about sub-goals, or as a basis for sub-policies or options. Universal Value Function Approximators (UVFAs) (Schaul et al., 2015) extend this idea by using a function approximator to parameterize a joint function of states and goal representations, allowing compact representation of an entire class of conditional value functions and generalization across classes of related goals.

While the above works assume a goal achievement reward to be available *a priori*, our work includes an approach to learning a reward function for goal achievement jointly with the policy. Several recent works have examined reward learning for goal achievement in the context of the Generative Adversarial Networks (GAN) paradigm (Goodfellow et al., 2014). The SPIRAL (Ganin et al., 2018) algorithm trains a goal conditioned policy with a reward function parameterized by a Wasserstein GAN (Arjovsky et al., 2017) discriminator. Similarly, AGILE (Bahdanau et al., 2018) learns an instruction-conditional policy where goals in a grid-world are specified in terms of predicates which should be satisfied, and a reward function is learned using a discriminator trained to distinguish states achieved by the policy from a dataset of instruction, goal state pairs.

Reward learning has also been used in the context of imitation. Ho & Ermon (2016) derives an adversarial network algorithm for imitation, while time-contrastive networks (Sermanet et al., 2017) leverage pre-trained ImageNet classifier representations to learn a reward function for robotics skills from video demonstrations, including robotic imitation of human poses. Universal Planning Networks (UPNs) (Srinivas et al., 2018) learn a state representation by training a differentiable planner to imitate expert trajectories. Experiments showed that once a UPN is trained the state representation it learned can be used to construct a reward function for visually specified goals. Bridging goal-conditioned

policy learning and imitation learning, Pathak et al. (2018) learns a goal-conditioned policy and a dynamics model with supervised learning without expert trajectories, and present zero-shot imitation of trajectories from a sequence of images of a desired task.

A closely related body of work to that of goal-conditioned reinforcement learning is that of unsupervised option or skill discovery. Machado & Bowling (2016) proposes a method based on an eigendecomposition of differences in features between successive states, further explored and extended in Machado et al. (2017). Variational Intrinsic Control (VIC) (Gregor et al., 2016) leverages the same lower bound on the mutual information as the present work in an unsupervised control setting, in the space of abstract options rather than explicit perceptual goals. VIC aims to jointly maximize the entropy of the set of options while making the options maximally distinguishable from their final states according to a parametric predictor. Recently, Eysenbach et al. (2018) showed that a special case of the VIC objective can scale to significantly more complex tasks and provide a useful basis for low-level control in a hierarchical reinforcement learning context.

Other work has explored learning policies in tandem with a task policy, where the task or environment rewards are assumed to be sparse. Florensa et al. (2017) propose a framework in which low-level skills are discovered in a pre-training phase of a hierarchial system based on simple-to-design proxy rewards, while Riedmiller et al. (2018) explore a suite of auxiliary tasks through simultaneous off-policy learning.

Several authors have explored a pre-training stage, sometimes paired with fine-tuning, based on unsupervised representation learning. Péré et al. (2018) and Laversanne-Finot et al. (2018) employ a two-stage framework wherein unsupervised representation learning is used to learn a model of the observations from which to sample goals for control in simple simulated environments. Nair et al. (2018) propose a similar approach in the context of model-free Q-learning applied to 3-dimensional simulations and robots. Goals for training the policy are sampled from the model's prior, and a reward function is derived from the latent codes. This contrasts with our non-parametric approach to selecting goals, as well as our method for learning the goal space online and jointly with the policy.

An important component of our method is a form of goal relabelling, introduced to the reinforcement learning literature as hindsight experience replay by Andrychowicz et al. (2017), based on the intuition that any trajectory constitutes a valid trajectory which achieves the goal specified by its own terminal observation. Earlier, Nair & Hinton (2006) employed a related scheme in the context of supervised learning of motor programs, where a program encoder is trained on pairs of trajectory realizations and programs obtained by expanding outwards from a pre-specified prototypical motor program through the addition of noise. Veeriah et al. (2018) expands upon hindsight replay and the all-goal update strategy proposed by Kaelbling (1993), generalizing the latter to non-tabular environments and exploring related strategies for skill discovery, unsupervised pre-training and auxiliary tasks. Levy et al. (2018) propose a hierarchical Q-learning system which employs hindsight replay both conventionally in the lower-level controller and at higher levels in the hierarchy. Nair et al. (2018) also employ a generalized goal relabeling scheme whereby the policy is trained based on a trajectory's achievement not just of its own terminal observation, but a variety of retrospectively considered possible goals.

## 6 EXPERIMENTS

We evaluate, both qualitatively and quantitatively, the ability of DISCERN to achieve visually-specified goals in three diverse domains – the Arcade Learning Environment (Bellemare et al., 2013), continuous control tasks in the DeepMind Control Suite (Tassa et al., 2018), and DeepMind Lab, a 3D first person environment (Beattie et al., 2016). Experimental details including architecture details, details of distributed training, and hyperparameters can be found in the Appendix. We compared DISCERN to several baseline methods for learning goal-conditioned policies:

**Conditioned Autoencoder (AE):** In order to specifically interrogate the role of the discriminative reward learning criterion, we replace the discriminative criterion for embedding learning with an $L^2$ reconstruction loss on $h_t$; that is, in addition to $\xi_\phi(\cdot)$, we learn an inverse mapping $\xi_\phi^{-1}(\cdot)$ with a separate set of parameters, and train both with the criterion $\|h_t - \xi_\phi^{-1}(\xi_\phi(h_t))\|^2$.

**Conditioned WGAN Discriminator:** We compare to an adversarial reward on the domains considered according to the protocol of Ganin et al. (2018), who successfully used a WGAN discriminator as a reward for training agents to perform inverse graphics tasks. The discriminator takes two pairs of images – (1) a *real* pair of goal images $(s_g, s_g)$ and (2) a *fake* pair consisting of the terminal state of the agent and the goal frame $(s_t, s_g)$. The output of the discriminator is used as the reward function for the policy. Unlike our DISCERN implementation and the conditioned autoencoder baseline, we train the WGAN discriminator as a separate convolutional network directly from pixels, as in previous work.

**Pixel distance reward (L2):** Finally, we directly compare to a reward based on $L^2$ distance in pixel space, equal to $\exp\left(-\|s_t - s_g\|^2 / \sigma_{\text{pixel}}\right)$ where $\sigma_{\text{pixel}}$ is a hyperparameter which we tuned on a per-environment basis.

All the baselines use the same goal-conditioned policy architecture as DISCERN. The baselines also used hindsight experience replay in the same way as DISCERN. They can therefore be seen as ablations of DISCERN's goal-achievement reward learning mechanism.

## 6.1 ATARI

The suite of 57 Atari games provided by the Arcade Learning Environment (Bellemare et al., 2013) is a widely used benchmark in the deep reinforcement learning literature. We compare DISCERN to other methods on the task of achieving visually specified goals on the games of Seaquest and Montezuma's Revenge. The relative simplicity of these domains makes it possible to handcraft a detector in order to localize the controllable aspects of the environment, namely the submarine in Seaquest and Panama Joe, the character controlled by the player in Montezuma's Revenge.

We evaluated the methods by running the learned goal policies on a fixed set of goals and measured the percentage of goals it was able to reach successfully. We evaluated both DISCERN and the baselines with two different goal buffer substitution strategies, *uniform* and *diverse*, which are described in the Appendix. A goal was deemed to be successfully achieved if the position of the avatar in the last frame was within 10% of the playable area of the position of the avatar in the goal for each controllable dimension. The controllable dimensions in Atari were considered to be the x- and y-coordinates of the avatar. The results are displayed in Figure 1a. DISCERN learned to achieve a large fraction of goals in both Seaquest and Montezuma's Revenge while none of the baselines learned to reliably achieve goals in either game. We hypothesize that the baselines failed to learn to control the avatars because their objectives are too closely tied to visual similarity. Figure 1b shows examples of goal achievement on Seaquest and Montezuma's Revenge. In Seaquest, DISCERN learned to match the position of the submarine in the goal image while ignoring the position of the fish, since the fish are not directly controllable. We have provided videos of the goal-conditioned policies learned by DISCERN on Seaquest and Montezuma's Revenge at the following anonymous URL `https://sites.google.com/view/discern-anonymous/home`.

## 6.2 DEEPMIND CONTROL SUITE TASKS

The DeepMind Control Suite (Tassa et al., 2018) is a suite of continuous control tasks built on the MuJoCo physics engine (Todorov et al., 2012). While most frequently used to evaluate agents which receive the underlying state variables as observations, we train our agents on pixel renderings of the scene using the default environment-specified camera, and do not directly observe the state variables.

Agents acting greedily with respect to a state-action value function require the ability to easily maximize $Q$ over the candidate actions. For ease of implementation, as well as comparison to other considered environments, we discretize the space of continuous actions to no more than 11 unique actions per environment (see Appendix A4.1).

The availability of an underlying representation of the physical state, while not used by the learner, provides a useful basis for comparison of achieved states to goals. We mask out state variables relating to entities in the scene not under the control of the agent; for example, the position of the target in the `reacher` or `manipulator` domains.

DISCERN is compared to the baselines on a fixed set of 100 goals with 20 trials for each goal. The goals are generated by acting randomly for 25 environment steps after initialization. In the case of

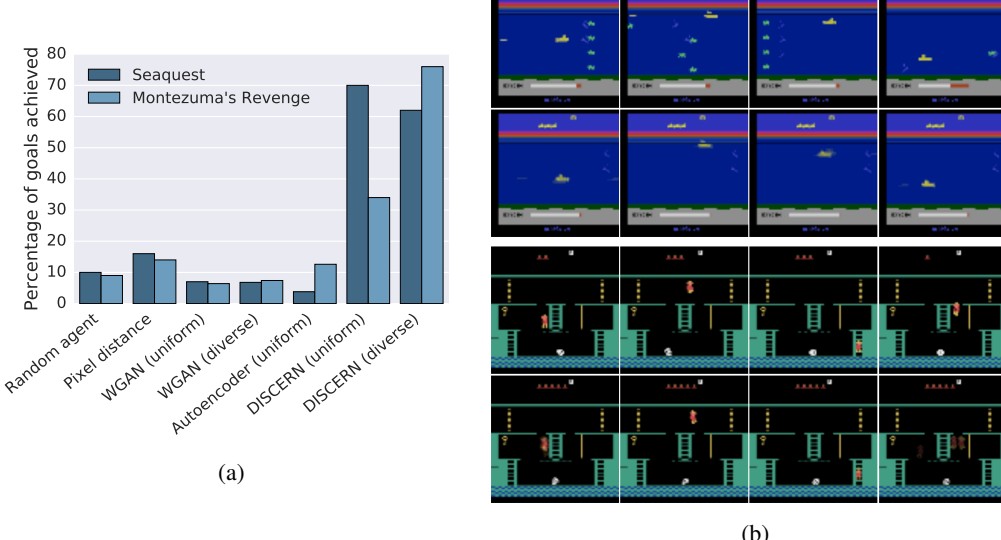

(a)

(b)

Figure 1: a) Percentage of goals successfully achieved on Seaquest and Montezuma's Revenge. b) Examples of goals achieved by DISCERN on the games of Seaquest (top) and Montezuma's Revenge (bottom). For each game, the four goal states are shown in the top row. Below each goal is the averaged (over 5 trials) final state achieved by the goal-conditioned policy learned by DISCERN after $T = 50$ steps for the goal above.

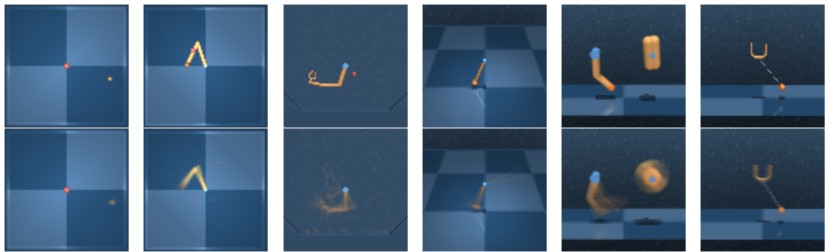

Figure 2: Average achieved frames for `point mass` (task `easy`), `reacher` (task `hard`), `manipulator` (task `bring ball`), `pendulum` (task `swingup`), `finger` (task `spin`) and `ball in cup` (task `catch`) environments. The goal is shown in the top row and the achieved frame is shown in the bottom row.

`cartpole`, we draw the goals from a random policy acting in the environment set to the `balance` task, where the pole is initialized upwards, in order to generate a more diverse set of goals against which to measure. Figure 3 compares learning progress of 5 independent seeds for the "uniform" goal replacement strategy (see Appendix A5 for results with "diverse" goal replacement) for 6 domains. We adopt the same definition of achievement as in Section 6.1. Figure 2 summarizes averaged goal achievement frames on these domains except for the cartpole domain for policies learned by DISCERN. Performance on cartpole is discussed in more detail in Figure 7 of the Appendix.

The results show that in aggregate, DISCERN outperforms baselines in terms of goal achievement on several, but not all, of the considered Control Suite domains. In order to obtain a more nuanced understanding of DISCERN's behaviour when compared with the baselines, we also examined achievement in terms of the individual dimensions of the controllable state. Figure 4 shows goal achievement separately for each dimension of the underlying state on four domains. The per-dimension results show that on difficult goal-achievement tasks such as those posed in `cartpole` (where most proposed goal states are unstable due to the effect of gravity) and `finger` (where a free-spinning piece is only indirectly controllable) DISCERN learns to reliably match the major dimensions of controllability such as the cart position and finger pose while ignoring the other

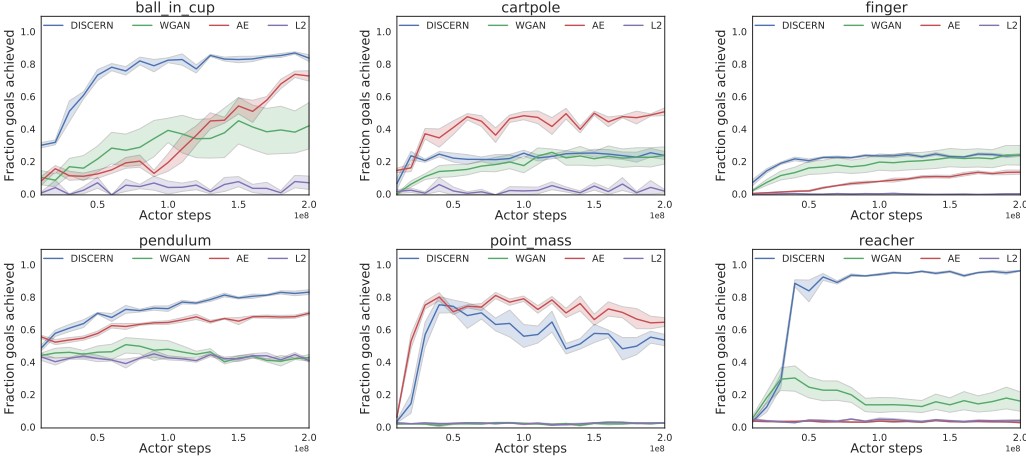

Figure 3: Quantitative evaluation of goal achievement on continuous control domains using the "uniform" goal substitution scheme (see Appendix A3). For each method, we show the fraction of goals achieved over a fixed goal set (100 images per domain).

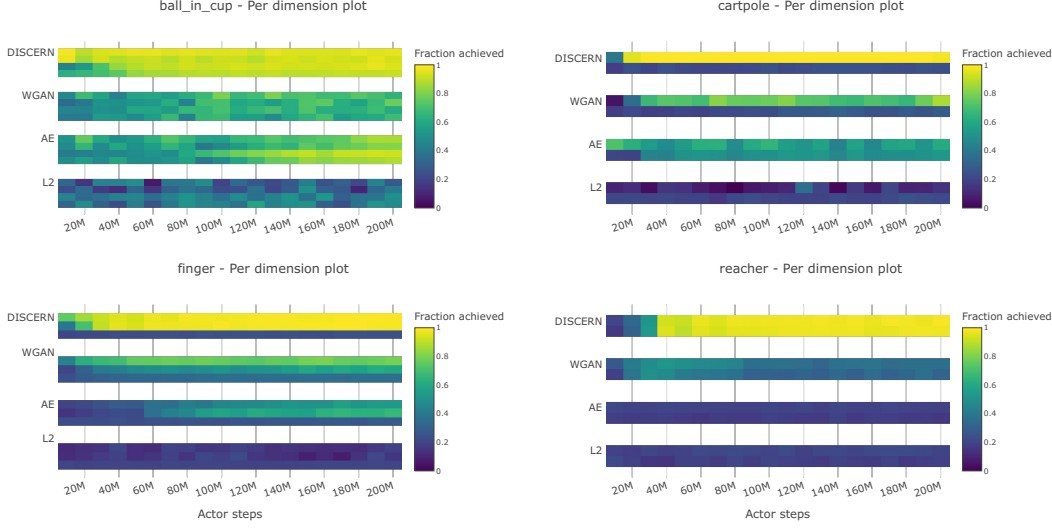

Figure 4: Per-dimension quantitative evaluation of goal achievement on continuous control domains using the "uniform" goal substitution scheme (Appendix A3). Each subplot corresponds to a domain, with each group of colored rows representing a method. Each individual row represents a dimension of the controllable state (such as a joint angle). The color of each cell indicates the fraction of goal states for which the method was able to match the ground truth value for that dimension to within 10% of the possible range. The position along the $x$-axis indicates the point in training in millions of frames. For example, on the reacher domain DISCERN learns to match both dimensions of the controllable state, but on the cartpole domain it learns to match the first dimension (cart position) but not the second dimension (pole angle).

dimensions, whereas none of the baselines learned to reliably match any of the controllable state dimensions on the difficult tasks cartpole and finger.

We omitted the manipulator domain from these figures as none of the methods under consideration achieved non-negligible goal achievement performance on this domain, how-ever a video showing the policy learned by DISCERN on this domain can be found at https://sites.google.com/view/discern-anonymous/home. The policy learned

Figure 5: Average achieved frames over 30 trials from a random initialization on the `rooms_watermaze` task. Goals are shown in the top row while the corresponding average achieved frames are in the bottom row.

on the `manipulator` domain shows that DISCERN was able to discover several major dimensions of controllability even on such a challenging task, as further evidenced by the per-dimension analysis on the `manipulator` domain in Figure 8 in the Appendix.

### 6.3 DEEPMIND LAB

DeepMind Lab (Beattie et al., 2016) is a platform for 3D first person reinforcement learning environments. We trained DISCERN on the `watermaze` level and found that it learned to approximately achieve the same wall and horizon position as in the goal image. While the agent did not learn to achieve the position and viewpoint shown in a goal image as one may have expected, it is encouraging that our approach learns a reasonable space of goals on a first-person 3D domain in addition to domains with third-person viewpoints like Atari and the DM Control Suite.

## 7 DISCUSSION

We have presented a system that can learn to achieve goals, specified in the form of observations from the environment, in a purely unsupervised fashion, i.e. without any extrinsic rewards or expert demonstrations. Integral to this system is a powerful and principled discriminative reward learning objective, which we have demonstrated can recover the dominant underlying degrees of controllability in a variety of visual domains.

In this work, we have adopted a fixed episode length of $T$ in the interest of simplicity and computational efficiency. This implicitly assumes not only that all sampled goals are approximately achievable in $T$ steps, but that the policy need not be concerned with finishing in less than the allotted number of steps. Both of these limitations could be addressed by considering schemes for early termination based on the embedding, though care must be taken not to deleteriously impact training by terminating episodes too early based on a poorly trained reward embedding. Relatedly, our goal selection strategy is agnostic to both the state of the environment at the commencement of the goal episode and the current skill profile of the policy, utilizing at most the content of the goal itself to drive the evolution of the goal buffer $\mathcal{G}$. We view it as highly encouraging that learning proceeds using such a naive goal selection strategy, however more sophisticated strategies, such as tracking and sampling from the frontier of currently achievable goals (Held et al., 2017), may yield substantial improvements.

DISCERN's ability to automatically discover controllable aspects of the observation space is a highly desirable property in the pursuit of robust low-level control. A natural next step is the incorporation of DISCERN into a deep hierarchical reinforcement learning setup (Vezhnevets et al., 2017; Levy et al., 2018; Nachum et al., 2018) where a meta-policy for proposing goals is learned after or in tandem with a low-level controller, i.e. by optimizing an extrinsic reward signal.

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

APPENDIX

A1    DISTRIBUTED TRAINING

We employ a distributed reinforcement learning architecture inspired by the IMPALA reinforcement learning architecture (Espeholt et al., 2018), with a centralized GPU learner batching parameter updates on experience collected by a large number of CPU-based parallel actors. While Espeholt et al. (2018) learns a stochastic policy through the use of an actor-critic architecture, we instead learn a goal-conditioned state-action value function with Q-learning. Each actor acts $\epsilon$-greedily with respect to a local copy of the $Q$ network, and sends observations $s_t$, actions $a_t$, rewards $r_t$ and discounts $\gamma_t$ for a trajectory to the learner. Following Horgan et al. (2018), we use a different value of $\epsilon$ for each actor, as this has been shown to improve exploration. The learner batches re-evaluation of the convolutional network and LSTM according to the action trajectories supplied and performs parameter updates, periodically broadcasting updated model parameters to the actors. As Q-learning is an off-policy algorithm, the experience traces sent to the learner can be used in the usual $n$-step Q-learning update without the need for an off-policy correction as in Espeholt et al. (2018). We also maintain actor-local replay buffers of previous actor trajectories and use them to perform both standard experience replay (Lin, 1993) and our variant of hindsight experience replay (Andrychowicz et al., 2017).

A2    ARCHITECTURE DETAILS

Our network architectures closely resemble those in Espeholt et al. (2018), with policy and value heads replaced with a $Q$-function. We apply the same convolutional network to both $s_t$ and $s_g$ and concatenate the final layer outputs. Note that the convolutional network outputs for $s_g$ need only be computed once per episode. We include a periodic representation $(\sin(2\pi t/T), \cos(2\pi t/T))$ of the current time step, with period equal to the goal length achievement period $T$, as an extra input to the network. The periodic representation is processed by a single hidden layer of rectified linear units and is concatenated with the visual representations fed to the LSTM. While not strictly necessary, we find that this allows the agent to become better at achieving goal states which may be unmaintainable due to their instability in the environment dynamics.

The output of the LSTM is the input to a dueling action-value output network (Wang et al., 2016). In all of our experiments, both branches of the dueling network are linear mappings. That is, given LSTM outputs $\psi_t$, we compute the action values for the current time step $t$ as

$$Q(a_t|\psi_t) = {\psi_t}^\mathsf{T}\mathbf{v} + \left( {\psi_t}^\mathsf{T}\mathbf{w}_{a_t} - \frac{1}{n}\sum_{a'_t} {\psi_t}^\mathsf{T}\mathbf{w}_{a'_t} \right) + b \qquad (6)$$

A3    GOAL BUFFER

We experimented with two strategies for updating the goal buffer. In the first strategy, which we call *uniform*, the current observation replaces a uniformly selected entry in the goal buffer with probability $p_{\text{replace}}$. The second strategy, which we refer to as *diverse* goal sampling attempts to maintain a goal buffer that more closely approximates the uniform distribution over all observation. In the diverse goal strategy, we consider the current observation for addition to the goal buffer with probability $p_{\text{replace}}$ at each step during acting. If the current observation $s$ is considered for addition to the goal buffer, then we select a random removal candidate $s_r$ by sampling uniformly from the goal buffer and replace it with $s$ if $s_r$ is closer to the rest of the goal buffer than $s$. If $s$ is closer to the rest of the goal buffer than $s_r$ then we still replace $s_r$ with $s$ with probability $p_{\text{add}-\text{non}-\text{diverse}}$. We used $L_2$ distance in pixel space for the diverse sampling strategy and found it to greatly increase the coverage of states in the goal buffer, especially early during training. This bears some relationship to Determinantal Point Processes (Kulesza et al., 2012), and goal-selection strategies with a more explicit theoretical foundation are a promising future direction.

A4    EXPERIMENTAL DETAILS

The following hyper-parameters were used in all of the experiments described in Section 6. All weight matrices are initialized using a standard truncated normal initializer, with the standard deviation

inversely proportional to the square root of the fan-in. We maintain a goal buffer of size $1024$ and use $p_{\text{replace}} = 10^{-3}$. We also use $p_{\text{add−non−diverse}} = 10^{-3}$. For the teacher, we choose $\xi_\phi(\cdot)$ to be an $\mathcal{L}_2$-normalized single layer of 32 tanh units, trained in all experiments with 4 decoys (and thus, according to our heuristic, $\beta$ equal to 5). For hindsight experience replay, a highsight goal is substituted $25\%$ of the time. These goals are chosen uniformly at random from the last 3 frames of the trajectory. Trajectories were set to be 50 steps long for Atari and DeepMind Lab and 100 for the DeepMind control suite. It is important to note that the environment was not reset after each trajectory, but rather the each new trajectory begins where the previous one ended. We train the agent and teacher jointly with RMSProp (Tieleman & Hinton, 2012) with a learning rate of $10^{-4}$. We follow the preprocessing protocol of Mnih et al. (2015), resizing to $84 \times 84$ pixels and scaling 8-bit pixel values to lie in the range $[0, 1]$. While originally designed for Atari, we apply this preprocessing pipeline across all environments used in this paper.

### A4.1 CONTROL SUITE

In the point_mass domain we use a control step equal to 5 times the task-specified default, i.e. the agent acts on every fifth environment step (Mnih et al., 2015). In all other Control Suite domains, we use the default. We use the "easy" version of the task where actuator semantics are fixed across environment episodes.

Discrete action spaces admit function approximators which simultaneously compute the action values for all possible actions, as popularized in Mnih et al. (2015). The action with maximal $Q$-value can thus be identified in time proportional to the cardinality of the action space. An enumeration of possible actions is no longer possible in the continuous setting. While approaches exist to enable continuous maximization in closed form (Gu et al., 2016), they come at the cost of greatly restricting the functional form of $Q$.

For ease of implementation, as well as comparison to other considered environments, we instead discretize the space of continuous actions. For all Control Suite environments considered except manipulator, we discretize an $A$-dimensional continuous action space into $3^A$ discrete actions, consisting of the Cartesian product over action dimensions with values in $\{−1, 0, 1\}$. In the case of manipulator, we adopt a "diagonal" discretization where each action consists of setting one actuator to $\pm 1$, and all other actuators to 0, with an additional action consisting of every actuator being set to 0. This is a reasonable choice for manipulator because any position can be achieved by a concatenation of actuator actions, which may not be true of more complex Control Suite environments such as humanoid, where the agent's body is subject to gravity and successful trajectories may require multi-joint actuation in a single control time step. The subset of the Control Suite considered in this work was chosen primarily such that the discretized action space would be of a reasonable size. We leave extensions to continuous domains to future work.

### A5 ADDITIONAL EXPERIMENTAL RESULTS

### A5.1 ATARI

We ran two additional baselines on Seaquest and Montezuma's Revenge, ablating our use of hindsight experience replay in opposite ways. One involved training the goal-conditioned policy only in hindsight, without any learned goal achievement reward, i.e. $p_{\text{HER}} = 1$. This approach achieved $12\%$ of goals on Seaquest and $11.4\%$ of goals on Montezuma's Revenge, making it comparable to a uniform random policy. This result underscores the importance of learning a goal achievement reward. The second baseline consisted of DISCERN learning a goal achievement reward *without* hindsight experience replay, i.e. $p_{\text{HER}} = 0$. This also performed poorly, achieving $11.4\%$ of goals on Seaquest and $8\%$ of goals on Montezuma's Revenge. Taken together, these preliminary results suggest that the combination of hindsight experience replay and a learned goal achievement reward is important.

### A5.2 CONTROL SUITE

For the sake of completeness, Figure 6 reports goal achievement curves on Control Suite domains using the "diverse" goal selection scheme.

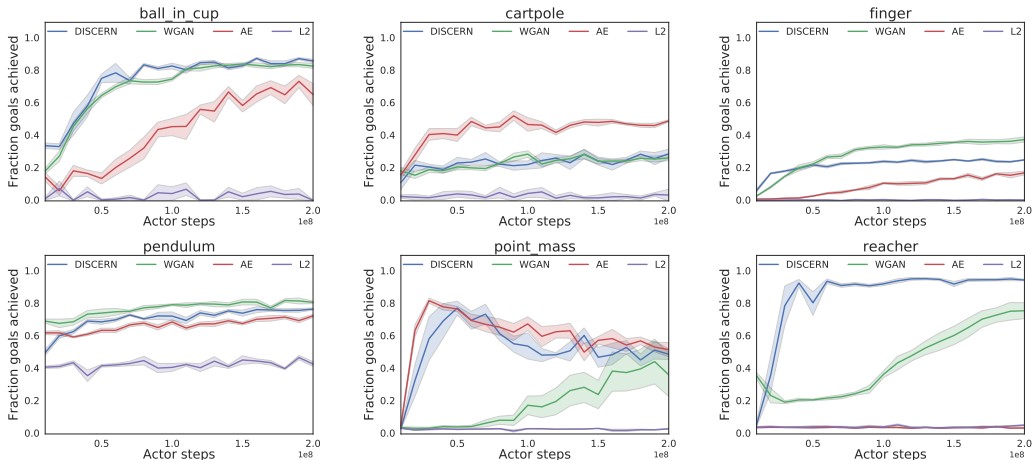

Figure 6: Results for Control Suite tasks using the "diverse" goal substitution scheme.

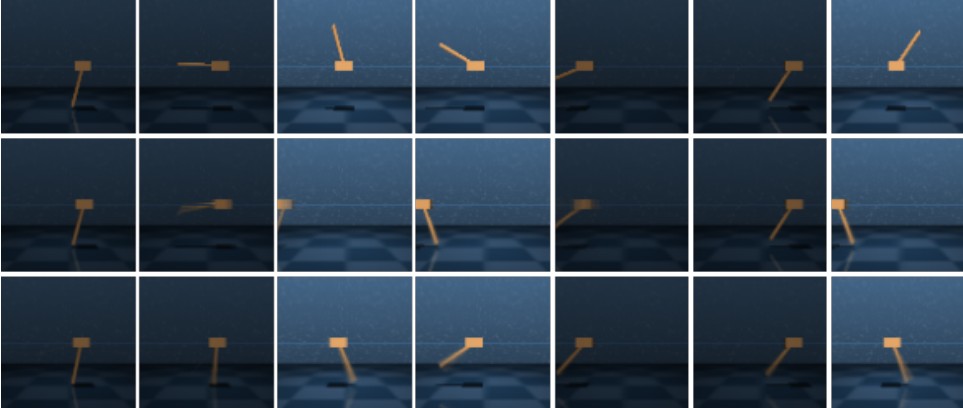

Figure 7: Average goal achievement on cartpole. Top row shows the goals. Middle row shows achievement by the Autoencoder baseline. Bottom row shows average goal achievement by DISCERN. Shading of columns is for emphasis. DISCERN always matches the cart position. The autoencoder baseline matches both cart and pole position when the pole is pointing down, but fails to match either when the pole is pointing up.

Figure 7 displays goal achievements for DISCERN and the Autoencoder baseline, highlighting DISCERN's preference for communicating with the cart position, and robustness to the pole positions unseen during training.

Figure 8: Per-dimension quantitative evaluation on the `manipulator` domain. See Figure 4 for a description of the visualization. DISCERN learns to reliably control more dimensions of the underlying state than any of the baselines.

