# OpenReview forum: "Unsupervised Control Through Non-Parametric Discriminative Rewards"
_ICLR.cc/2019/Conference_

### Official Review · AnonReviewer3 · 2018-10-23
**A strong paper with innovative ideas, but somewhat unclear methods and results**

**Rating:** 7
**Confidence:** 3

**Review:**


In this paper, the authors address the problem of learning to achieve perceptually specified goals in a fully unsupervised way. For doing so, they simultaneously learn a goal-conditioned policy and a goal achievement reward function based on the mutual information between goals sampled from an a priori distribution and states achieved using the goal-conditioned policy. These two learning processes are coupled through the mutual information criterion, which seems to result in efficient state representation learning for the visual specified goal space. A key feature is that the resulting metrics in the visual goal space helps the agent focus on what it can control and ignore distractors, which is critical for open-ended learning.

Overall, the idea looks very original and promissing, but the methods are quite difficult to understand under the current form, the messages from the results are not always clear, and the lack of ablative studies makes it difficult to determine which of the mechanisms are crucial in the system performance and which are not.

* Clarification of the methods:

Given the key features outlined above, I believe the work described in this paper has a lot of potential, but the main issue is that the methods are not easy to get, and the authors could do a better job in that respect. Here is a list of remarks meant to help the authors write a clearer presentation of their method:

- the "problem formulation" section contains various things. Part of it could be inserted as a subsection in Section 3, and the last paragraph may rather come into the related work section.

- in Section 3, optimization paragraph, the details given after "As will be discussed"... might rather come in Section 4 were most of all other details are given.

- in Section 4, I would refer to Algorithm 1 only in the end of the section after all the details have been explained: I went first to the algorithm and could not understand many details that are explained only afterwards.

- in Algorithm 1, shouldn't the two procedures be called "Imitator" and "Teacher", rather than "actor" and "learner", to be consistent with the end of Section 3?

- there must be a mathematical relationship between $\xsi_\phi$ and $\hat{q}$, but I could not find this relationship anywhere in the text. What is $\xsi_\phi$ is never introduced clearly...

- p4: we treat h as fixed ... => explain why.

- I don't have a strong background about variational methods, and it is unclear to me why using an expanding set of goals corresponding to already seen states recorded in a buffer makes it that maximizing the log likelihood given in (4) is easier than something else.

More generally, the above are local remarks from a reader who did not succeed in getting a clear picture of what is done exactly and why. Anything you can do to give a more didactic account of the methods is welcome.

* Related work:

The related work section is too poor for a strong paper like this one. Learning to reach goals and learning goal representations are two extremely active domains at the moment and the authors should position themselves with respect to more of these works. Here is a short list in which the authors may find many more relevant papers:

 (Machado and Bowling, 2016), (Machado et al., 2017), GoalGANs (Florensa et al., 2018), RIG (Nair et al., 2018), Many-Goals RL (Veeriah et al., 2018), DAYN (Eysenbach et al., 2018), FUN (Vezhnevets et al., 2017), HierQ, HAC (Levy et al., 2018), HIRO (Nachum et al., 2018), IMGEP (Pere et al., 2018), MUGL IMGEP (Laversanne-Finot et al., 2018).

It would also be useful to position yourself with respect to Sermanet et al. : "Unsupervised Perceptual Rewards for Imitation Learning".

About state representation learning, if you consider the topic as relevant for your work, you might have a look at the recent survey from Lesort et al. (2018).

External comments on ICLR web site also point to missing references. The authors should definitely consider doing a much more serious job in positioning their work with respect to the relevant literature.

* Experimental study:

The algorithm comes with a lot of mechanisms and small tricks (at the end of Section 3 and in Section 4) whose importance is never assessed by specific experimental studies. This matters all the more than some of the details do not seem to be much principled. It would be nice to have elements to figure out how important they are with ablative studies putting them aside and comparing performance. Among other things, I would be glad to know how well the system performs without its HER component. Is it critical?

The same about the goal sampling strategy, as mentioned in the discussion: how critical is it in the performance of the algorithms?

- Fig. 1b is not so easy to exploit: it is hard to figure out what the reader should actually extract from these figures

- difficult tasks like cartpole: other papers mention cartpole as a rather easy task.

In the begining of Section 4, the authors mention that the mechanisms of DISCERN naturally induce a form of curriculum (which may be debated), but this aspect is not highlighted clearly enough in the experimental study.

In my opinion, studying fewer environments but giving a more detailed analysis of the performance of DISCERN and its variations in these environment would make the paper stronger.



* typos:

p3: the problem (of) learning a goal achievement reward function

In (3), p_g should most probably be p_{goal}

p4: we treated h(.) ... and did not adapt => treat, do not

p9: needn't => need not

---

> ### Author Response · Authors · 2018-11-22
> **Response to AR3 (1/2)**
>
> We thank AnonReviewer3 for their thoughtful review. We are currently preparing a revised manuscript which will address notational issues, typos as well as an expanded related work section. We address other specific comments below.
>
> > and the lack of ablative studies makes it difficult to determine which of the mechanisms are crucial in the system performance and which are not.
>
> We have ablated the reward function learner in 3 ways: first, by keeping everything fixed but swapping the discriminative objective for an autoencoding objective. Second, by swapping the reward learner for a separate network trained using the criterion from Ganin et al (again, keeping the agent architecture fixed; we also use the same proportion of hindsight relabeled trajectories, a point not stressed but which will be made in a revision). Third, using a reward based on a fixed notion of visual similarity in terms of L2 distance, where we tuned the bandwidth hyperparameter of this baseline to make it as strong as possible. If there are specific ablations AR3 would like to see we can attempt to address them.
>
> > - in Section 4, I would refer to Algorithm 1 only in the end of the section after all the details have been explained: I went first to the algorithm and could not understand many details that are explained only afterwards.
>
> We agree and will do this.
>
> > - in Algorithm 1, shouldn't the two procedures be called "Imitator" and "Teacher", rather than "actor" and "learner", to be consistent with the end of Section 3?
>
> The algorithm box is explained in terms of an experience-gathering procedure (Actor) and parameter update procedure (Learner) that is a split independent of the specifics introduced by DISCERN (see, e.g. [1] and [2] for other examples of such an exposition). Each of these procedures makes use of (and in the case of the learner, trains) both of the conceptual pieces (“imitator” and “teacher”, as you say). We chose this conceptual breakdown of the algorithm for the pseudocode block as it closely reflects our parallel distributed implementation (similar to Espeholt et al (2018)). Although a serial (or even more directly a single machine, multi-process) implementation is straightforward to derive from this conceptual partitioning, the reverse is not true, and so we felt it more valuable to provide the Actor/Learner description.
>
> [1] https://surreal.stanford.edu/img/surreal-corl2018.pdf
> [2] https://arxiv.org/abs/1803.00933
>
> > - there must be a mathematical relationship between $\xsi_\phi$ and $\hat{q}$, but I could not find this relationship anywhere in the text. What is $\xsi_\phi$ is never introduced clearly…
>
> We have introduced e() as the composition of h() and the learned embedding xi(), and use e() in equation 4. We introduced e() specifically to reduce clutter but we see now that this has caused more confusion. Several reviewers have commented on the lack of clarity here so we will address this in a revision later this week.
>
> > - p4: we treat h as fixed ... => explain why.
>
> We re-use the same convolutional net for computational efficiency, i.e. in order to avoid the need to learn a separate convolutional network. We will add explanatory text to the revision. Note that this is a common procedure in deep actor-critic methods, where the convolutional network features of the policy are often reused for the critic without backpropagating the critic’s gradients into the shared features (see, e.g. the “Learning from pixels” results in Section 6 of Tassa et al, 2018); we will expand upon this in the text. We experimented with optimizing the convolutional network with respect to both the reward learning loss and the reinforcement learning loss and found it to perform worse in practice than only optimizing it with respect to the RL loss. Joint optimization would likely require careful tuning of a weighting hyperparameter trading off the two losses.

---

> > ### Comment · AnonReviewer3 · 2018-11-23
> > **Missing revised paper**
> >
> > Well, most of the reponses look satisfactory, but now we need to see how they will be implemented into the revised version of the paper... The deadline is close now!

---

> ### Author Response · Authors · 2018-11-22
> **Response to AR3 (2/2)**
>
> > - I don't have a strong background about variational methods, and it is unclear to me why using an expanding set of goals corresponding to already seen states recorded in a buffer makes it that maximizing the log likelihood given in (4) is easier than something else.
>
> Equations 2 and 3 make reference to q(s_g|s_T) which, in its most general form, would be a conditional distribution responsible for assigning a scalar density to each point s_g giving the likelihood that a trajectory (from goal_conditional policy pi) which achieved terminal state s_T was in fact “intending” (i.e. given the conditioning goal) s_g. This is potentially a very difficult density modeling problem and it is unclear that such a distribution would be efficiently learnable, especially online in tandem with the policy. We replace this with a classification problem between K+1 candidates for the goal, as classification is generally regarded as easier than density modeling. Furthermore our use of a non-parametric matching network-style objective allows the classifier to perform a different classification "task" every time (the terminal observation of the goal episode is always different, as well as the set of candidate goals) while nonetheless generalizing across classification instances.
>
> Using an expanding set of goals is motivated by the fact that we do not assume to have access to the set of all possible goals, and rely on exploration through our behaviour policy. Thus evolving the goal buffer over time allows us to train on newly encountered states as goals.
>
> > The algorithm comes with a lot of mechanisms and small tricks (at the end of Section 3 and in Section 4) whose importance is never assessed by specific experimental studies. This matters all the more than some of the details do not seem to be much principled. It would be nice to have elements to figure out how important they are with ablative studies putting them aside and comparing performance.
>
> Our initial experiments used q-hat(s_g|s_t) as the reward but this has the potential to introduce noise through decoy sampling, reducing the realized reward when a decoy happens to be close to the goal. We found that simply using the rectified embedding cosine similarity worked better, but we do not believe this is necessarily an optimal choice.
>
> > Among other things, I would be glad to know how well the system performs without its HER component. Is it critical?
>
> We performed additional experiments on Atari which we will include in the appendix. It appears that on the two Atari domains considered, performance drops to approximately 25-30% goal achievement in both instances without Hindsight Experience Replay. This could possibly be improved by choosing a different function of the embedding similarity as the reward.
>
> > The same about the goal sampling strategy, as mentioned in the discussion: how critical is it in the performance of the algorithms?
>
> We report results with both sampling strategies on Control Suite and Atari tasks. Diverse sampling seems to be important on Montezuma’s Revenge (where exploration is more difficult) but otherwise both strategies seem to perform comparably well.
>
> > - difficult tasks like cartpole: other papers mention cartpole as a rather easy task.
>
> Here we refer to the cartpole task as we pose it, i.e. achieving a specific cart and pole position at a specific time (the end of the goal episode). Unlike the standard balancing task it is impossible to maintain an arbitrary pole position due to the effects of gravity. We will make this clearer in the text.
>
> > In the begining of Section 4, the authors mention that the mechanisms of DISCERN naturally induce a form of curriculum (which may be debated), but this aspect is not highlighted clearly enough in the experimental study.
>
> We are unsure how we would go about highlighting this and welcome your suggestions.

---

### Official Review · AnonReviewer1 · 2018-11-02

**Rating:** 5
**Confidence:** 5

**Review:**

The paper proposes an unsupervised learning algorithm to learn a goal conditioned policy and the corresponding reward function (for the goal conditioned policy) by maximizing the mutual information b/w the goal state and the state achieved by running  the goal conditioned policy for K time steps. The paper proposes a tractable way to maximize this mutual information objective, which basically amounts to learning a reward function for the goal conditioned policy.

The paper is very well written and easy to understand.

MISSING CITATIONS: Original UVFA [1] paper should be cited while citing goal conditioned policies.

In the paragraph,  "Goal distribution" , the paper uses a non parametric approach to approximate the goal distribution. Previous works ([2], [3]) have used such an approach and relevant work should be cited.

[1] http://proceedings.mlr.press/v37/schaul15.html
[2] Many Goals Reinforcement Learning https://arxiv.org/abs/1806.09605
[3] Recall Traces: Backtracking Models for efficient RL https://arxiv.org/abs/1804.00379

I wonder if  learning the variational distribution would be tricky in scenarios where one need to extract a representation of the end state that can distinguish states based on actions required to reach them. Like consider a U-shaped maze
|       |         |
|       |         |
|_A__|__B__|
In this maze, even though the states represented by points A and B close to each other, but functionally they are very far apart.  I'm curious as to what authors have to say in this regard.

Baseline Comparison: I find the experiment results not really convincing. First, comparison to other "unsupervised" exploration methods like Variational information maximizing exploration (VIME),  Variational Intrinsic Control (VIC), Curiosity driven learning (using inverse models) is missing.  I understand that VIME and VIC are really not scalable as compared to the proposed method, and hence it should be easy to construct a toy task where it is possible to intuitively understand whats really going on, as well as one can compare with the other baselines (VIME, VIC).

I would recommend authors to study a toyish environment in a proper way as compared to running (incomplete) experiments on 3 different set of envs. It would make the paper really strong.

---

> ### Author Response · Authors · 2018-11-22
> **Response to AR1**
>
> We thank the reviewer for the careful review of our work and your comments. As noted in replies to AR2, we are pushing a revision shortly to address concerns raised, chiefly a new Related Work section.
>
> > MISSING CITATIONS: Original UVFA [1] paper should be cited while citing goal conditioned policies.
>
> This was an oversight (the citation was there but got cut in editing), and will be addressed in a revision later this week.
>
> > In the paragraph,  "Goal distribution" , the paper uses a non parametric approach to approximate the goal distribution. Previous works ([2], [3]) have used such an approach and relevant work should be cited.
>
> > [1] http://proceedings.mlr.press/v37/schaul15.html
> > [2] Many Goals Reinforcement Learning https://arxiv.org/abs/1806.09605
> > [3] Recall Traces: Backtracking Models for efficient RL https://arxiv.org/abs/1804.00379
>
> We cite HER, of which [2] is an extension. The idea of progress-based prioritization is unlikely to work in our context, as the notion of goal-completion is highly non-stationary. However, we agree that non-parametric buffers have a rich history in the context of deep reinforcement learning, so we’ve added a citation in our next revision to the paper by Lin that introduced them.
>
> The backtracking procedure in [3] is completely orthogonal to our work. The prioritization scheme for what states to start backtracking from relies entirely on extrinsic reward, which we do not use and to which we do not presume access..
>
> > I wonder if  learning the variational distribution would be tricky in scenarios where one need to extract a representation of the end state that can distinguish states based on actions required to reach them. Like consider a U-shaped maze
> > |       |         |
> > |       |         |
> > |_A__|__B__|
> > In this maze, even though the states represented by points A and B close to each other, but functionally they are very far apart.  I'm curious as to what authors have to say in this regard.
>
> We believe this is precisely the reason to learn a state embedding rather than rely on naive or predetermined notions of similarity. In the point mass task, for example, pixel renderings of any two non-overlapping positions of the point mass will have equal L2 distance from one another. A reward based on this will not reward positions nearby to the goal differently from far away positions. The situation becomes even more complex when the goal and the observation contain differences outside of the agent’s control.
>
> > Baseline Comparison: I find the experiment results not really convincing. First, comparison to other "unsupervised" exploration methods like Variational information maximizing exploration (VIME),  Variational Intrinsic Control (VIC), Curiosity driven learning (using inverse models) is missing.  I understand that VIME and VIC are really not scalable as compared to the proposed method, and hence it should be easy to construct a toy task where it is possible to intuitively understand whats really going on, as well as one can compare with the other baselines (VIME, VIC).
>
> We are aware of VIME but as far as we can tell, every condition examined in that work uses an externally defined reward function, which we do not. Furthermore, VIME is a strategy for improving exploration while DISCERN is a method for learning to achieve visually specified goals. It is not clear how they could be compared.
>
> Variational Intrinsic Control has not been shown to work in the setting of goals specified as visual observations and the results presented in that work on complex visual environments are very preliminary. We agree that applying VIC to this problem directly is an interesting direction but consider it out of scope for the present work.
>
> Regarding curiosity-driven models: while these can be learned in an unsupervised way and there is a strong connection to the original formulation of empowerment (in the one-step case), after a significant review we have yet to come across a paper where state-conditioned goal achievement could be done without significant algorithmic modifications.
>
> > I would recommend authors to study a toyish environment in a proper way as compared to running (incomplete) experiments on 3 different set of envs. It would make the paper really strong.
>
> We believe that several of the tasks considered (reacher, point mass) are among the simplest instantiations of the problem we consider that are still interesting (i.e. high-dimensional pixel observations). The reason we chose to study both Atari and the Control Suite tasks in-depth is because they represent very different characteristics and are externally defined. We agree with Reviewer 2’s assessment that tasks in the deep RL literature are too often cherry-picked and we wanted to demonstrate breadth of applicability, which we believe we have.

---

> > ### Comment · AnonReviewer1 · 2018-11-25
> > **Thanks for your reply.**
> >
> > "The backtracking procedure in [3] is completely orthogonal to our work. The prioritization scheme for what states to start backtracking from relies entirely on extrinsic reward, which we do not use and to which we do not presume access.. "  "Using an expanding set of goals is motivated by the fact that we do not assume to have access to the set of all possible goals, and rely on exploration through our behaviour policy. Thus evolving the goal buffer over time allows us to train on newly encountered states as goals. "
> >
> > I understand the motivation behind your work, as well as backtracking procedure. While reading your paper, backtracking procedure work came to my mind, as there motivation was also the same in using an expanding set of goals,  which worked well for them. So, just wanted to point it out.
> >
> > "We believe this is precisely the reason to learn a state embedding rather than rely on naive or predetermined notions of similarity. In the point mass task, for example, pixel renderings of any two non-overlapping positions of the point mass will have equal L2 distance from one another. A reward based on this will not reward positions nearby to the goal differently from far away positions. The situation becomes even more complex when the goal and the observation contain differences outside of the agent’s control."
> >
> > I'm not sure, if I understand why for your method it would treat the 2 states differently.
> >
> > "Regarding curiosity-driven models: while these can be learned in an unsupervised way and there is a strong connection to the original formulation of empowerment (in the one-step case), after a significant review we have yet to come across a paper where state-conditioned goal achievement could be done without significant algorithmic modifications."
> >
> > Again, I agree with authors. I was more of the impression, that just giving the visual goal as a conditional input in the formulation of (Pathak et. al).

---

> > > ### Author Response · Authors · 2018-11-28
> > > **Maze example**
> > >
> > > Regarding the example of a U-shaped maze, if the goal images for two states are indistinguishable, then we would expect our approach to either pick one of the goals or go to the one that is closer to the agent’s current position. We agree that this is a limitation of using a single observation to specify a goal, but it applies equally to other approaches relying on this formulation. One way to work around this without changing the goal formulation is to provide the agent with a sequence of waypoint goals one after the other, leading it to the desired states. This is the approach used in “Zero-Shot Visual Imitation” by Pathak et al.

---

### Official Review · AnonReviewer2 · 2018-11-02
**Important problem, reasonable initial attempt, room for improvement**

**Rating:** 8
**Confidence:** 5

**Review:**

Summary:

The authors take up an important problem in unsupervised deep reinforcement learning which is to learn perceptual reward functions for goal-conditioned policies without extrinsic rewards from the environment. The problem is important in order to push the field forward to learning representations of the environment without predicting value functions from scalar rewards and learn more generalizable aspects of the environment (the authors call this mastery) as opposed to just memorizing the best sequence of actions in typical value/policy networks.

Model-based methods are currently hard to execute as far as mastery is concerned and goal-conditioned value functions are a good alternative. The authors, therefore, propose to learn UVFA (Schaul et al) with a learned perceptual reward function r(s, s_g) where 's' and 's_g' are current and goal observations respectively. They investigate a few choices for deriving this reward, such as pixel-space L2 distance, Auto-Encoder feature space, WGAN Discriminator (as done in SPIRAL - Ganin and Kulkarni et al), and their approach: cosine similarity based log-likelihood for similarity metric (as in Matching Networks).  They show that their approach works better than other alternatives on a number of visual goal-based tasks.

Specific aspects:

1. A slight negative: I find the whole pipeline extremely hacky and raises serious questions on whether this paper/technique is easy to apply on a wide variety of tasks. It gives me the suspicion that the environments were cherry-picked for showing the success of the proposed method, though, that's, in general, true of most deep RL papers. It would be nice if the authors instead wrote the paper from the perspective of proposing a new benchmark (it would be amazing if the benchmark is open sourced so that it will lead to more people working specifically on this setting and a lot more comparisons).

-- Revision: The pipeline is hacky, but getting GAN based reward learning to work is also not very straightforward. The authors do plan to release the detectors used for the benchmarking.

2. To elaborate on the above, these are the portions I find hacky:
(i) Need for decoy observations to learn an approximate log-likelihood
(ii) Using sparse reward for all transitions except the final terminal state: Yes, I am aware of the fact that HER has already shown sparse rewards are easier to learn value functions with, compared to dense rewards. But I am genuinely surprised that you have pretty much the same setting (ie re-label only terminal transition, r(s_T, s_g)) and motivate the need for learning a perceptual metric. If the information bits per transition is similar to HER in terms of the policy network's objective function, I am not sure why you need to learn a perceptual reward then? There's also no baseline comparison with just naive HER on image observations. That will be worth seeing actually. I feel this kind of comparisons are more interesting and important for the message of the paper. Note that in other papers cited in this, such as SPIRAL, UPN, etc, the reward metrics are used for every state transition.
(iii) In addition to naive image HER, I would really like to see a SPIRAL + HER baseline as is. ie use the GAN reward for all transitions and also use relabeling for successes. My prior belief is that this will work really well. I would really like to know how the reward for each transition in the trajectory works (both for SPIRAL and your approach) and how the naive HER works.

--Revision: The authors have added HER baselines. Agreed with the authors that comparison of per-timestep perceptual reward vs terminal state perceptual reward is a good topic for future work.

3. Another place I really found confusing throughout the paper is the careless swapping of notations, especially in the xi(h) and e(h). Please use consistent notations especially in equation (3), the pseudocode and the rest of the paper.

4.  a. Would be nice to know if a VAE feature space metric is bad, but not a strict requirement if you don't have time to do it. But I think showing Euclidean metric baseline on VAE is better than an AE.
      b. Another baseline that is related is to learn a metric with a triplet loss as in Sermanet's work. Or any noise contrastive loss approach (such as CPC). The matching networks approach is similar in spirit. Just pointing out as reference and something worth trying, but not expecting it to be done for rebuttal.

5. Overall, I think this is a good paper, gives a good overview of an important problem; the matching networks idea is nice and simple; but the paper could be more broader in terms of writing than trying to portray the success of DISCERN specifically. I would be happy accepting it even if the SPIRAL baseline or VAE / AE baseline worked as well as the matching networks because I think those approaches are more principled and likely to require fewer hacks and could be applied to a lot of domains easily. I also hope the authors run the baselines I asked for just to make the paper more scientifically complete.

6. Not a big deal for me in terms of deciding acceptance, but for the sake of good principles in academics, related work could be stronger, though I can understand it must have been small purely due to page limits.

Some papers which could be cited are (1) Unsupervised Perceptual Rewards (though it uses AlexNet pre-trained), (2) Time Contrastive Networks (which also uses AlexNet and doesn't really work on single-view tasks but is a good citation to add), (3) Original UVFA  (definitely has to be there given you even use the abbreviation for the keywords description of the paper)

7. Some slightly incorrect facts/wording in the paper: The two papers cited in model-based methods (Oh and Chiappa) are not really unsupervised. They use a ton of demonstrations to learn those world models. Better citation might be David Ha's World Models or Chelsea Finn's Video Prediction.

---

> ### Author Response · Authors · 2018-11-22
> **Response to AR2 (1/3)**
>
> We thank AnonReviewer2 for their careful reading of our paper and their valuable feedback.
>
> We have heard concerns regarding related work from several reviewers, and are currently finishing a complete rewrite of this section, and will publish a revision in the next few days which addresses this and several other concerns raised. We respond to specific concerns below.
>
> > 1. A slight negative: I find the whole pipeline extremely hacky and raises serious questions on whether this paper/technique is easy to apply on a wide variety of tasks. It gives me the suspicion that the environments were cherry-picked for showing the success of the proposed method, though, that's, in general, true of most deep RL papers.
>
> We evaluated on examples of three families of visual domains that have very little in common and demonstrated success to various degrees on all of them. We know of no other work which evaluates on continuous control from pixels and Atari games in the same paper. Our main criterion for selection among Control Suite tasks was the dimensionality of the action space (and therefore the cardinality of the discretized action space; we will clarify this in the text), which concerns a limitation of Q learning rather than our method built on top of Q learning. We'd also note that DISCERN is not uniformly the winner on our "whole state" goal achievement metric on the Control Suite tasks; if we had wanted to cherry pick, including these would be an odd choice.
>
> > (it would be amazing if the benchmark is open sourced so that it will lead to more people working specifically on this setting and a lot more comparisons).
>
> The domains we used are already open source. We plan on open-sourcing the detectors we used for the Atari task as well as the code we used to extract ground truth from the Control Suite environments, in order to enable comparison.
>
> > (i) Need for decoy observations to learn an approximate log-likelihood
>
> We disagree that using decoys is hacky. Methods like noise contrastive estimation rely on a similar mechanism and are a standard way of doing approximate maximum likelihood training. What we propose is a non-parametric formulation of mutual information maximization which we further instantiate approximately by sampling. We note that contrastive predictive coding, concurrent work with our own which you mention in your review, also employs negative examples or decoys.
>
> > (ii) Using sparse reward for all transitions except the final terminal state: Yes, I am aware of the fact that HER has already shown sparse rewards are easier to learn value functions with, compared to dense rewards. But I am genuinely surprised that you have pretty much the same setting (ie re-label only terminal transition, r(s_T, s_g)) and motivate the need for learning a perceptual metric. If the information bits per transition is similar to HER in terms of the policy network's objective function, I am not sure why you need to learn a perceptual reward then? There's also no baseline comparison with just naive HER on image observations. That will be worth seeing actually.
>
> We attempted training purely by HER on the Atari tasks in the way you suggest. This did not work well and the percentage of goals achieved was worse than for a random agent on both Seaquest and Montezuma’s Revenge. We will add these results to the appendix.

---

> ### Author Response · Authors · 2018-11-22
> **Response to AR2 (2/3)**
>
> > Note that in other papers cited in this, such as SPIRAL, UPN, etc, the reward metrics are used for every state transition.
>
> We consulted the SPIRAL paper and found that it uses the same reward scheme as us, i.e. a single non-zero reward on the final step of the episode. As you mentioned earlier, the HER paper makes a good case for why sparse rewards could be better than per-step rewards.
>
> > (iii) In addition to naive image HER, I would really like to see a SPIRAL + HER baseline as is.
>
> Thank you for pointing this out. We give both the autoencoder and the WGAN-trained policy the benefit of hindsight experience replay as well. Hence, the WGAN baseline in our paper corresponds to exactly what you suggest. We will relabel the WGAN and AE baselines as WGAN+HER and AE+HER to make this clearer. As you can see, WGAN+HER works well on many Control Suite tasks, but does not work on Atari, where moving distractor objects are present.
>
> > I would really like to know how the reward for each transition in the trajectory works (both for SPIRAL and your approach) and how the naive HER works.
>
> We experimented early on with dense rewards and found it to work worse, possibly due to tradeoffs between partial achievement early and more complete achievement later. We consider dense rewards to be an important future direction but we haven’t included this comparison as the original SPIRAL work used a sparse reward, and we don’t believe we have the space to give this topic the treatment it deserves.
>
> > 3. Another place I really found confusing throughout the paper is the careless swapping of notations, especially in the xi(h) and e(h). Please use consistent notations especially in equation (3), the pseudocode and the rest of the paper.
>
> We will attempt to address this in a revision posted later this week. We intended e() to be the composition of h(), xi(), and an L2 normalization and introduced it to reduce clutter. We apologize if it made things hard to follow.
>
> > 4.  a. Would be nice to know if a VAE feature space metric is bad, but not a strict requirement if you don't have time to do it. But I think showing Euclidean metric baseline on VAE is better than an AE.
>
> We did experiment with both VAE and AE baselines. The AE baseline performed better so we omitted the VAE baseline in order to avoid crowding the figures.
>
> >  b. Another baseline that is related is to learn a metric with a triplet loss as in Sermanet's work.
>
> As far as we can tell, neither of the Sermanet et al papers are directly comparable because the reward function relies on demonstrations of “good” trajectories, rather than an intrinsically desired end state.

---

> ### Author Response · Authors · 2018-11-22
> **Response to AR2 (3/3)**
>
> > 5. Overall, I think this is a good paper, gives a good overview of an important problem; the matching networks idea is nice and simple; but the paper could be more broader in terms of writing than trying to portray the success of DISCERN specifically. I would be happy accepting it even if the SPIRAL baseline or VAE / AE baseline worked as well as the matching networks because I think those approaches are more principled and likely to require fewer hacks and could be applied to a lot of domains easily.
>
> We disagree with the assertion that SPIRAL/AE/VAE baselines are more principled. Both our reward learner and agent approximately optimize the same objective, whereas density modeling or reconstruction objectives for reward learning are in fact introducing a secondary objective unrelated to the reinforcement learning problem at hand.
>
> GANs are somewhat notorious for being difficult to train. Discriminator-based rewards can also degrade when the generator’s performance becomes such that the discriminator has little or no basis for telling real from synthetic (which is perhaps not a problem for SPIRAL as the faces are not sufficiently realistic reproductions, but see e.g. Bahdanau et al, 2018’s AGILE method for a discussion). Our cooperative objective does not seem to suffer from these degeneracies, and the agent performing well does not have the potential to negatively impact the reward learner’s learning dynamics.
>
> Finally, we view DISCERN’s contribution as a method for robustly operationalizing mutual information in the case of goal-based RL. None of the baseline methods would have any reason to learn a similarity metric that ignores distracting elements outside the agent’s control, and indeed this is validated by their stronger performance on visually simpler Control Suite tasks and degraded performance on Atari domains, where important elements of the observed game state (namely enemies) cannot be reliably matched exactly.
>
> > I also hope the authors run the baselines I asked for just to make the paper more scientifically complete.
>
> As we note above, we have run the hindsight-only baseline for Atari and will be adding to the Appendix. We believe this is the harder of the two quantitatively evaluated domain families for a hindsight-only setup to cope with due to the presence of uncontrollable distracting elements of the state, and we have verified that indeed it is unable to achieve a significant fraction of goals. We can run these baselines for the Control Suite task as well if requested but we don’t believe this will be as informative as the Atari result.
>
> > (2) Time Contrastive Networks (which also uses AlexNet and doesn't really work on single-view tasks but is a good citation to add),
>
> We agree, and in fact already cite this work.
>
> > (3) Original UVFA  (definitely has to be there given you even use the abbreviation for the keywords description of the paper)
>
> This is indeed an oversight, UVFA was mentioned in a previous version and the citation was cut in an edit and never reintroduced. We will correct this.
>
> > 7. Some slightly incorrect facts/wording in the paper: The two papers cited in model-based methods (Oh and Chiappa) are not really unsupervised. They use a ton of demonstrations to learn those world models.
>
> While these papers do use pretrained agents to collect the training data, the model-learning algorithm is unsupervised and could be used on data from a random policy without modification. This is likely to reduce their performance, but our point is that even with this "cheat", performing model-based RL on them doesn't work.
>
> > Better citation might be David Ha's World Models or Chelsea Finn's Video Prediction.
>
> The “World Models” setup is more inline with our own, compared to the papers we initially cited, so we will cite World Models in our upcoming revision. But since their results are only on two environments, one of which is a rather peculiarly constructed task (bullet avoidance in VizDoom), we should regard them as preliminary.
>
> Video Prediction: Similarly to the papers we do cite, it uses training data that isn't random (generated by a human I believe). It is farther from our domains of interest and isn't evaluated in terms of model-based RL.

---

> ### Comment · AnonReviewer2 · 2018-11-22
> **Response to Authors' Feedback**
>
> I have read the detailed feedback of the authors and appreciate their effort in addressing my comments by running additional experiments and also explaining why certain aspects of proposed experiments could warrant a future research topic.
>
> The authors are right in saying that GAN based methods such as SPIRAL are not anyway less hacky than noise contrastive methods given the level of trickery / skill needed to make those methods train in a stable way. So, I take back my comment on that.
>
> This paper is important to encourage more work on RL without rewards, an emerging field. So, I strongly recommend accepting the paper and updating my score to 8.

---

### Public Comment · (anonymous) · 2018-10-01
**missing reference to DFP?**

"We have presented a system that can learn to achieve goals, specified in the form of observations
from the environment" - paper

"Assuming that the goal can be expressed in terms of future measurements,"  from "Learning to Act by Predicting the Future", ICLR 2017 https://arxiv.org/abs/1611.01779

While the approaches are quite different, but the main idea is close enough to mention or even tested against DFP, given the strong performance of the latter

---

> ### Author Response · Authors · 2018-10-02
> **Thank you for the feedback**
>
> Thank you for the feedback! The quoted line is a bit ambiguous in this context. Unlike the linked paper, our work doesn’t have a low-dimensional “measurement” observation stream. Rather, when we refer to goals specified as observations, we mean the full observation given to the agent (i.e. the image). This lack of a measurement channel in the environments tested precludes having "Learning to Act by Predicting the Future" as a baseline.
>
> We agree that DFP is related since it provides an alternative specification of goals by using additional low-dimensional “measurements”. We did not cite the paper because our related work section focuses on methods for achieving visually specified goals. This is already a large area to cover and including alternative goal specification methods such as DFP, language-based goals, and demonstrations is beyond the scope of our work.

---

### Comment · AnonReviewer2 · 2018-10-19
**Clarification**

Dear authors,

Can you please add the visual results on the supplementary site for all the benchmarks in the paper? ie other DM Control tasks and the DM Lab tasks. It would also be really useful and good for clarity if you can explicitly point out the goal image observation as a separate image observation rather than superimposing it in the video. I would like to clearly know what part of the Atari screen is fed in as is, and what is not. Example, it seems like the scores are blurred in Seaquest. And the DM Control task is confusing to me, in terms of whether the small ball's goal position is part of the goal or not, or is it just the pose matching of the arm.

---

> ### Author Response · Authors · 2018-10-30
> **Clarifications**
>
> Thank you for your comment. We are not sure if including additional results is permitted outside of the rebuttal phase so we will add additional visualizations to the supplementary site once the rebuttal period begins. Please see below for detailed answers and clarifications.
>
> >  It would also be really useful and good for clarity if you can explicitly point out the goal image observation as a separate image observation rather than superimposing it in the video.
>
> Thank you for the suggestion. The videos are included in addition to Figures 1 and 2, which show the goal frames provided to the agent on Atari and Control Suite tasks. We found that superimposing the goal on the videos made it easier to judge how closely the agent matches the goal. However, we will include some videos where the goal is not superimposed as you suggest.
>
> >  I would like to clearly know what part of the Atari screen is fed in as is
>
> The entire RGB frame for both the goal and the current time-step (84x84, preprocessed according to the protocol in Mnih et al (2015)) is fed in.
>
> > Example, it seems like the scores are blurred in Seaquest.
>
> We are not altering the observation or the goal in any way beyond the standard preprocessing above. Downsampling to 84x84 does impact legibility of the score, and we are displaying (in both the paper and the videos) the downsampled observations. The achievement frames in the manuscript (bottom row of Figures 1b and 2), are averages over the final frames of trajectories started with the same goal but different initial states, so some blurring will naturally occur; however, all frames seen by the agent are unaltered beyond the fixed preprocessing noted above.
>
>
> > And the DM Control task is confusing to me, in terms of whether the small ball's goal position is part of the goal or not, or is it just the pose matching of the arm.
>
> We feed in entire frames as goals, so the small yellow ball in the “manipulator” task is indeed part of the goal (although the ball is not visible in some frames if it’s falling or bouncing). As the video for the “manipulator task” shows, DISCERN learns to approximately match the position of the arm in the goal image, but largely ignores the ball. The results in the DeepMind Control Suite paper show [1] that the “manipulator” task is difficult when using pixel observations as we do; the state of the art D4PG agent was not able to solve the “manipulator” task from pixels when given an extrinsic reward for moving the ball to a target location. Our setting is even more difficult since the agent does not receive such an extrinsic reward.
>
> The “manipulator” domain also includes a pink marker which represents the target location for the ball for the extrinsic reward task. The location of the pink marker is chosen randomly by the environment at the beginning of each environment episode; hence agent has no control over the location of the marker. Because we are not using the extrinsic reward, the pink marker is a distracting object similar to the skull in Montezuma’s Revenge. DISCERN is however robust to these distracting elements and learns to ignore them while matching aspects of the environment which are under its control. Similarly, the “reacher” and “point_mass” domains also include a pink marker that is not controllable by the agent.
>
> We will clarify these aspects of the environment in the appendix and update the paper during the rebuttal phase.
>
> [1] “DeepMind Control Suite” Tassa et al. - https://arxiv.org/abs/1801.00690

---

### Author Response · Authors · 2018-11-24
**Related work revised, another revision on the way**

Dear reviewers,

As the Related Work section was a point of contention for more than one reviewer, and the most significant revision required, we felt it prudent to address it first and as soon as possible. We have posted a revised copy with an entirely rewritten Related Work section incorporating your feedback.

We will post another revision tomorrow which addresses the rest of the textual concerns.

---

### Author Response · Authors · 2018-11-27
**Another revision posted**

Dear reviewers,

We have posted another revision addressing more textual concerns.

- We have added a description in-text regarding our use of HER for the reward learning baselines. We intended to relabel figures but were unable to, we will relabel them in the final camera-ready.
- We have moved the Algorithm box forward in the text as suggested by AnonReviewer3, as well as addressed all typos they pointed out.
- We have clarified in text regarding why cartpole is "difficult" in our setting.
- We have added references to suggested deep HRL works to the discussion, where we mention HRL.
- We have added additional notes to the appendix regarding why we picked the Control Suite environments we did.
- We have added a citation to Ha & Schmidhuber's "World Models" as suggested by AnonReviewer2.
- We have cited Lin's experience replay work when we introduce the time-evolving non-parametric  buffer of previous states that we use as a source of goals.
- Finally, Appendix A5.1 now includes a description of HER ablation results on Atari, i.e. training without HER and training only on HER.

We thank the reviewers for their help in improving the paper.

---

### Comment · AnonReviewer2 · 2018-12-11
**Suggestion for a correction**

Authors,

When reading the paper again, I came across a line in related work which is potentially incorrect:
"Recently, Eysenbach et al. (2018) showed that a special case of the VIC objective can scale to significantly more complex tasks and provide a useful basis for low-level control in a hierarchical reinforcement learning context."

It's hard to make a clear comparison between the nature of tasks considered in the two papers. The tasks picked by Eysenbach et al are actually quite easy if the goal is to just see diverse random behaviors for locomotion as long as appropriate action limits are set for the controller and random rewards are optimized. Operating from pixels is a different challenge altogether. "Scale" and "Significantly more complex tasks" are loose and vague statements. I am assuming this paper will be well read and received as far as goal-based / unsupervised RL are concerned and it's important related work is covered without any misleading interpretations.

---

> ### Author Response · Authors · 2018-12-13
> **Will revise for camera-ready**
>
> Greetings R2,
>
> We agree that this is potentially misleading, as VIC did indeed show preliminary results on domains with pixel observations. We will replace "scale" with "be applied" and "significantly more complex tasks" with "simulated continuous control tasks" in order to be more concise (please let us know if you'd suggest an alternative wording).
>
> Regards,
>
> -- Paper 1256 Authors

---

> > ### Comment · AnonReviewer3 · 2018-12-16
> > **More about the relationship to DIAYN**
> >
> > When thinking more about it (a late thought, sorry), the "discrimination" idea in this paper is quite close in spirit to the idea in "Diversity is All You Need" (Eysenbach et al. 2018, which is about to get accepted at ICLR 2019)).
> > At first glance, the main differences are in the mathematical way to put it, and the fact that DISCERN deals with "distractor" goals which cannot be under the control of the agent.
> >
> > In the final version of this paper, I would be glad to know more about this relationship: is there a simple connection that can be established between the mathematical frameworks? Could the authors compare how both frameworks (DIAYN and DISCERN) behave in two simple benchmarks, one without a distractor and one with some distractors? Are there other important differences which should be put forward?

---

> > > ### Comment · AnonReviewer2 · 2018-12-18
> > > **Discussion should be about VIC more than DIAYN**
> > >
> > > @Reviewer 3,
> > >
> > > This paper is more about a reasonable simplification + CPC-like MI information estimator of Variational Intrinsic Control (VIC). I don't particularly see the need to discuss the relationship with DIAYN given that DIAYN is itself a trivial simplification of VIC while the authors clearly derive their idea from VIC by (1) replacing options with a goal image and making the policy goal-conditioned, (2) the entropy term of available options reduces to entropy of goal states which is not a parametric function of the agent's policy or the MI estimator between goal and final state; (3) MI between option and final state = MI between goal observation and final observation - which is approximated by the authors with a contrastive loss with associative embeddings.
> > >
> > > There are other nice details in this approach:
> > >
> > > The authors train a separate embedding on top of the conv features of the image observations for the contrastive loss but don't back prop to the conv features used by the policy which optimizes the mutual information objective - which is a good stable implementation.

---

### Comment · AnonReviewer3 · 2019-01-15
**Still some issues**

I'm currently reading the paper and found a few points that deserve clarification.

* In Section 4, the relationship between $e()$ and $\xhi_\phi$ is not explicitly written. The authors probably mean $e(s) = \xi_\phi(h(s))^T\xi_\phi(h(s))$.

If this is so, rather than defining $l_g$ as they do, they could rewrite (5) as

$$... = log \frac{exp(\beta e(s_g))}{exp(\beta e(s_g)) + \sum^K exp(e(d_k))}$$.

* In Section 4 the authors say that Q is trained with $Q(\lambda)$, but in Appendix A2 they describe something more complicated related to IMPALA and using an LSTM. Where is the truth?

I would be glad to see these points fixed in the final version of the paper (or the arxiv one)

---

> ### Author Response · Authors · 2019-01-15
> **Re: Still some issues**
>
>
> We thank the reviewer for pointing out the notational mistake, wherein e() was maintained in several in-text instances even after it was purged from the equations. We will correct this in the final version.
>
> Regarding the confusion about Q(lambda) and the IMPALA architecture, we draw distinctions between the type of policy (deterministic vs. stochastic), the reinforcement learning algorithm, the data-gathering strategy, and the particular architecture of function approximators used to represent the policy.
>
> We train a greedy, deterministic agent which represents its Q function using a feedforward network that drives a recurrent memory. While the extension of Q-learning agents to the recurrent case is straightforward (especially in the case of relatively short episodes such as ours, where the LSTM can be fully unrolled in time), we note now that it was previously explored by Hausknecht & Stone (2015). We will cite this work in the final version for the sake of fair attribution and added clarity.
>
> There are two orthogonal similarities between the agents used in our experiments and IMPALA. The first is that we gather experience for our replay buffer in a distributed manner with a centralized learner, resembling IMPALA’s approach at a high level. The second is with regards to the function approximators chosen: our network architectures are identical to those employed in the IMPALA work, except that rather than policy and state-value output layers, an output layer based on equation (6) computes action-values (the previous reward and action are also omitted as inputs). Q(lambda) is then used to compute the targets for this layer.
>
> We stress that the choice of experience-gathering strategy, the network architecture (including the use of an LSTM), and even the use of Q(lambda) targets are implementation choices that are not central to DISCERN’s contribution.

---

### Meta-Review · Area_Chair1 · 2018-12-13
**Concerned about the rigor of experiments**

**Confidence:** 5
**Recommendation:** Accept (Poster)

**Metareview:**

This paper introduces an unsupervised algorithm to learn a goal-conditioned policy and the reward function by formulating a mutual information maximization problem. The idea is interesting, but the experimental studies seem not rigorous enough. In the final version, I would like to see some more detailed analysis of the results obtained by the baselines (pixel approaches), as well as careful discussion on the relationship with other related work, such as Variational Intrinsic Control.